# The efficacy and safety of sodium tanshinone IIA sulfonate injection in the treatment of unstable angina pectoris: A systematic review and meta-analysis

**Xiaoqi Wu** [1], **Maoxia Fan** [1], **Sining Wei** [2]*, **Dong Guo** [1]*

**1** Shandong University of Traditional Chinese Medicine, Jinan, Shandong Province, China, **2** Affiliated Hospital of Shandong University of Traditional Chinese Medicine, Jinan, Shandong Province, China

☯ These authors contributed equally to this work.

* 71001817@sdutcm.edu.cn (SW); qlyp@sdutcm.edu.cn (DG)

## Abstract

### Objective

To systematically evaluate the efficacy and safety of Sodium tanshinone IIA sulfonate injection (STS) in the treatment of unstable angina pectoris (UAP).

### Methods

CNKI, Wanfang, VIP, CBM, PubMed, Cochrane Library, Web of Science, Embase were searched by computer. The research covers the clinical randomized controlled trials of STS in the treatment of unstable angina pectoris published from the establishment of the library to January 31, 2023. Two researchers independently screened the literature, extracted data and evaluated the risk of research bias, and then conducted meta-analysis with RevMan5.3 software.

### Results

A total of 37 randomized controlled trials were included, involving 3926 patients in total. Meta analysis results showed that, compared with conventional western medicine alone, STS combined with conventional western medicine could reduce the frequency (SMD = -2.61, 95%CI[-4.27, -0.96], $P$ = 0.002) and duration (SMD = -4.01, 95%CI[-6.18, -1.84], $P$ = 0.0003) of angina pectoris, improve ECG efficacy (OR = 3.61, 95%CI[2.79, 4.68], $P$<0.00001) and clinical symptom efficacy (OR = 4.02, 95%CI[3.32, 4.87], $P$<0.00001), reduce TG(SMD = -0.60, 95%CI[-1.04, -0.16], $P$ = 0.008), TC(SMD = -3.86, 95%CI[-6.37, -1.34], $P$ = 0.003), and LDL-C(SMD = -1.54, 95%CI[-2.67, -0.42], $P$ = 0.007), decrease plasma viscosity(SMD = -1.02, 95%CI[-1.58, -0.47], $P$<0.0003), whole blood low shear viscosity(SMD = -0.85, 95%CI[-1.21, -0.49], $P$<0.00001), whole blood high shear viscosity (SMD = -0.82, 95%CI[-1.44, -0.20], $P$ = 0.009), and erythrocyte aggregation index(SMD = -1.00, 95%CI[-1.75, -0.25], $P$ = 0.009), and bring down CRP(SMD = -1.39, 95%CI[-1.91, -0.86], $P$<0.00001). The incidence of adverse reactions in the treatment group was higher

**Funding:** This study was funded/supported by the National Academic Schools of Traditional Chinese Medicine,(XSLP-2013-35).

**Competing interests:** The authors have declared that no competing interests exist.

than that in the control group (OR = 2.26, 95%CI[1.06, 4.85], $P$ = 0.04). Neither of the two groups suffered from abnormal liver and kidney function during the study process.

## Conclusion

STS combined with routine treatment has a definite clinical efficacy and certain safety in the treatment of UAP, but it needs to be further confirmed by high-quality and low-bias randomized controlled trials in the future.

## 1. Introduction

Unstable angina pectoris (UAP) is a common type of angina pectoris, it is caused by coronary artery stenosis, increased blood viscosity, and atheromatous plaque blocking the blood vessels. The main manifestation is that the anterior chest area suffers from repeated pain with increasing frequency and severity. With the development of the disease, it can cause acute myocardial infarction [1,2]. UAP is currently one of the common chronic diseases that seriously affect the health and quality of patients' life, and modern medicine has greatly improved the treatment of this disease. Conventional methods include anticoagulation, antiplatelet aggregation drugs β-receptor blockers, nitrates, angiotensin converting enzyme inhibitors, statins and other drugs [3], but they come with problems such as prominent side effects and drug dependence.

According to its clinical symptoms and signs, Traditional Chinese Medicine (TCM) classifies UAP as "chest pain", "heartache", and "angina pectoris". In recent years, with the continuous development of TCM preparation technology, more and more TCM preparations have been used to treat unstable angina pectoris, and their efficacy and safety have also received widespread attention [4]. There have been extensive pharmacological research on Salvia miltiorrhiza, and its chemical components are mainly divided into fat-soluble component tanshinone and water-soluble component salvianolic acid. The most widely studied compound in modern times is tanshinone. Sodium tanshinone IIA sulfonate is a substance obtained by chemical modification of the monomer tanshinone IIA extracted from Salvia miltiorrhiza [5], which is a Chinese herbal extract found in China to treat coronary heart disease. With very good anti-myocardial ischemia and hypoxia effect, sodium tanshinone IIA sodium sulfonate (STS) can inhibit angiotensin, reduce the occurrence of myocardial remodeling, and improve the clinical symptoms of angina pectoris [6]. STS is administered by intravenous drip, which is more conducive to maintaining drug concentration, improving blood circulation, expanding coronary arteries, and reducing heart rate [7].

There have been frequent reports of randomized controlled clinical trial (RCTs) on the treatment of UAP with STS, but the quality of these studies is low and the conclusions are not completely consistent. Using the method of evidence-based medicine, the effectiveness and safety of STS in the treatment of unstable angina pectoris were systematically evaluated, in an effort to provide strong evidence support for clinical research and application.

## 2. Material and methods

The methodology of this study followed the Cochrane manual and the report outlined in this study complied with the PRISMA2020 checklist [8]. This system evaluation has been registered on the PROSPERO website with the registration number: PROSPERO CRD42022369967.

## 2.1 Search strategy

The retrieval was conducted from multiple databases such as CNKI, Wanfang, VIP, CBM, PubMed, Cochrane Library, Web of Science, and Embase, covering RCTs of STS in the treatment of UAP from the establishment of the database to the publication on January 31, 2023. The Chinese search terms included: "coronary heart disease", "angina pectoris", "unstable angina pectoris", "chest pain", "true heartache", "tanshinone IIA sodium sulfonate injection", "tanshinone injection", "randomized", and "randomized controlled trial"; and the English search terms include: "Unstable angina", "Angina pectoris", "Sodium tanshinone IIA sulfonate", "Tanshinone IIA sulfonate", "randomized controlled trial", and "randomized". The search strategy of combining subject word with free word was adopted, and adjustment was made where necessary according to different retrieval strategies of each database.

## 2.2 Inclusion criteria

**2.2.1 The type of research.** STS combined with routine treatment for RCTs of UAP;

**2.2.2 Research object.** All the patients were in compliance with the diagnostic criteria for unstable angina in the "Guidelines for Emergency Diagnosis and Treatment of Acute Coronary Syndrome (2019)", regardless of age, gender, course of disease, and other conditions.

**2.2.3 Intervention study.** The control group was treated with routine treatment (RT), including regular rest, oxygen intake, nitric acid ester, β-receptor antagonists, antiplatelet agglutinating drugs, angiotensin converting enzyme inhibitors, calcium antagonists, and statins for treatment, and so on. The treatment group was treated with STS combined with the routine treatment.

**2.2.4 Outcome indicators.** The main outcome indicators are electrocardiographic efficacy (ECG), frequency of angina attacks, and duration of angina attacks. The secondary outcome indicators are efficacy of clinical symptoms (total effective = significant + effective), blood rheology (plasma viscosity, whole blood low viscosity, whole blood high shear viscosity; red cell aggregation index), blood lipid [total cholesterol (TC), triacylgycerol (TG), low-density lipoprotein cholesterol (LDL-c), high-density lipoprotein cholesterol (HDL-c)], C-reaction protein (CRP), and adverse reactions.

## 2.3 Exclusion criteria

1. Non-clinical randomized controlled trial;

2. The interventions fails to meet the inclusion criteria, the diagnosis is unclear, or there exists other serious diseases;

3. The data is not rigorous or the effective outcome indicator information cannot be extracted;

4. Repeated publications, reviews, and animal experiments.

## 2.4 Data management and analysis

The two researchers (MXF and XQW) independently read and selected the full text of the literature and extracted relevant information, including the basic information of the literature (author, publication time), the number of patients, intervention methods (intervention measures, dose, duration), outcome indicators, adverse reactions, and other specific content. Cross-check was performed and any differences that arises was referred to a third party (SNW) for discussion and evaluation, and information was extracted from the final literature.

The included literature was evaluated based on the bias risk assessment tool for clinical randomized controlled trials in the Cochrane manual. The evaluation items involved random sequence generation methods, allocation concealment, blinding implementation of subjects and investigators, blinding implementation of outcome evaluators, incomplete outcome reports, selective reports, and other sources of bias. Each of the above items was rated as "low risk", "high risk", and "unclear".

## 2.5 Statistical analysis

Statistical analysis was conducted using RevMan5.3 software provided by the Cochrane system. The odds ratio (OR) was used as the combined statistic for counting data, and the mean difference (MD) was used as the combined statistic for measuring data. If different measurement methods or units were used, the standardized mean difference (SMD) was used as the combined statistic for measuring data, all expressed as 95% confidence interval (CI). The test for heterogenicity can be quantitatively judged by I2 and P values.When $P > 0.1$ and $I^2 \leq 50\%$, the fixed effect model was used for meta analysis, otherwise the random effect model was used for combined effect quantity analysis. At the same time, subgroup analysis was used to explore the source of heterogeneity in outcome indicators with large heterogeneity, and then sensitivity analysis or descriptive analysis was conducted through a one-by-one exclusion method. When the number of included studies was greater than 10, a funnel map was drawn to further assess whether there was publication bias.

# 3. Results

## 3.1 Results of the literature search

A total of 849 articles were retrieved, 37 of which were finally included after screening [9–45], and 36 were written in Chinese and 1 in English. 3926 patients were involved, including 2161 patients in the treatment group and 1765 in the control group. The literature screening process is shown in Fig 1, and the basic information for the included studies is shown in Table 1.

## 3.2 Quality evaluation of literature

The Cochrane manual was used to assess the bias of the included literature. A total of 37 [9–45] articles were included, where 2 studies [17,27] used a random number table method, 1 study [28] used a computer-generated program randomization method, 1 study [25] mentioned random grouping according to the order of patient visits, 3 studies [13,36,39] mentioned random grouping according to the order of admission, and the remaining 30 studies all mentioned random grouping but failed to give specific description to the randomization methods. Besides, 7 studies [13,17,25,27,28,36,39] identified allocation considerations as high risk, 2 studies [10,45] mentioned the use of single blinding, and the specific assessment information of bias risk included in various studies was shown in Fig 2.

## 3.3 Meta analytic result

### 3.3.1 Main outcome indicators.

(1)ECG efficacy:

19 studies [9,11,14,20,21,24,26,30,32–34,36,38–41,43–45] reported electrocardiographic efficacy, and heterogeneity testing showed that there was little heterogeneity between the studies ($P = 0.91$, $I^2 = 0$). The fixed effect model was used for meta-analysis, and the results showed that the difference was statistically significant (OR = 3.61, 95% CI [2.79, 4.68], P < 0.00001),

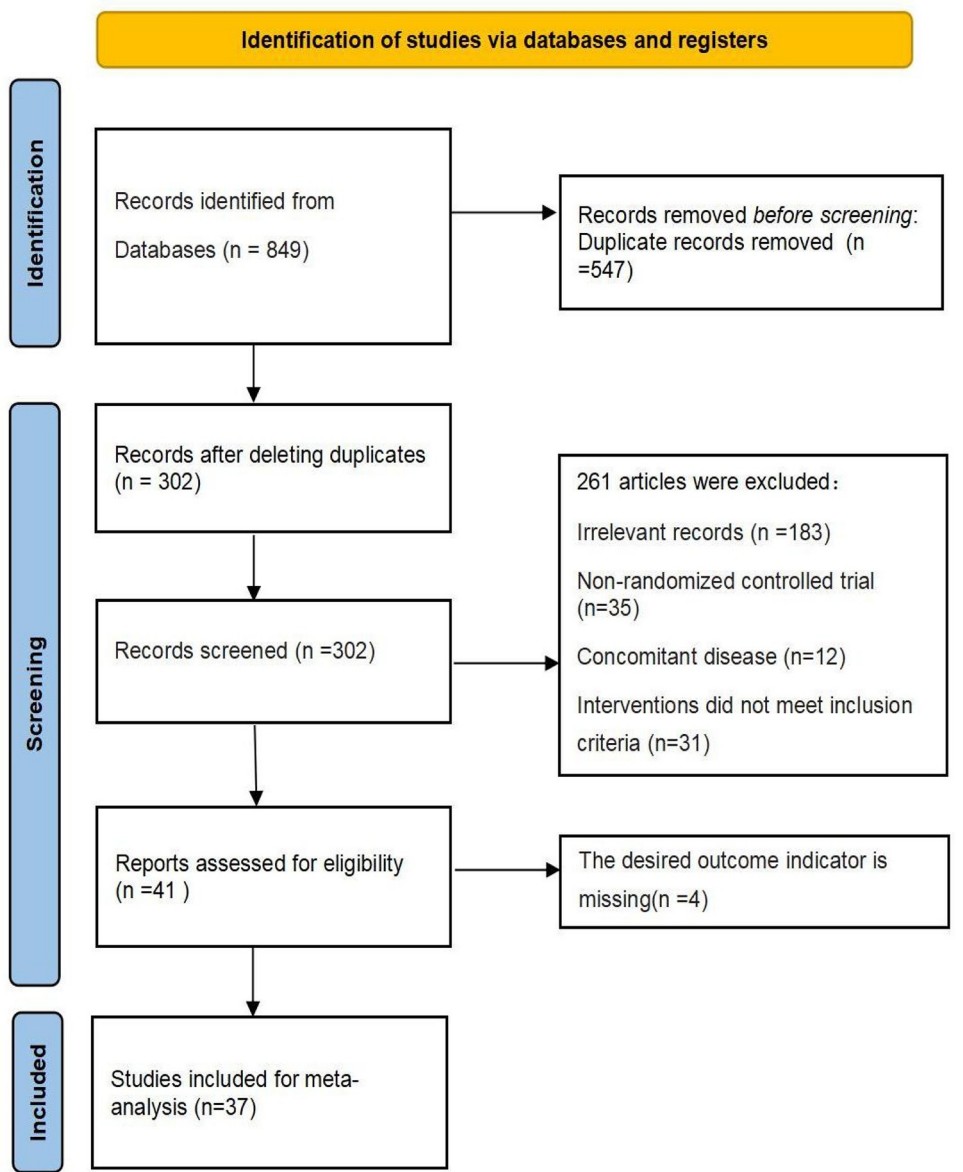

**Fig 1. PRISMA flow-chart of the study selection process.**

indicating that the treatment group received better efficacy in electrocardiogram improvement than the control group, as shown in Fig 3.

(2)Frequency of angina pectoris attacks:

7 studies [9,10,13,16,22,26,40] reported the frequency of angina pectoris attacks, and the heterogeneity test showed that there was significant heterogeneity among the studies ($P < 0.00001$, $I^2 = 99\%$). Through subgroup analysis, sensitivity analysis, and full-text reading, no source of significant heterogeneity was found through factors such as dose and course of treatment. Therefore, a randomized response model was used for meta analysis, and the results showed that the difference was statistically significant (SMD = -2.61, 95% CI [-4.27, -0.96], P = 0.002), indicating that the treatment group received better efficacy in improving the frequency of angina attacks than the control group, as shown in Fig 4.

**Table 1. Basic characteristic of include studies.**

| References | Group | Sample size | Interventions | Duration | outcome indicators | adverse reactions |
|---|---|---|---|---|---|---|
| Zhang Y 2013 [9] | T | 83 | RT+STS 60mg qd | 14D | ①②③④⑤ | Yes |
| | C | 80 | RT | | | |
| Li RJ 2017 [10] | T | 40 | RT+STS 80mg qd | 14D | ①② | No |
| | C | 40 | RT | | | |
| Li PQ 2012 [11] | T | 40 | RT+STS 70mg qd | 14D | ①④ | No |
| | C | 40 | RT | | | |
| Xue Y 2012 [12] | T | 40 | RT+STS 50mg qd | 14D | ① | No |
| | C | 40 | RT | | | |
| Jin ZG 2016 [13] | T | 50 | RT+STS 60mg qd | 14D | ①②⑥ | Unclear |
| | C | 50 | RT | | | |
| Ju H 2013 [14] | T | 30 | RT+STS 60mg qd | 10D | ①④ | Yes |
| | C | 30 | RT | | | |
| Liu ZH 2014 [15] | T | 45 | RT+STS 60mg qd | 14D | ① | Unclear |
| | C | 45 | RT | | | |
| Si QX 2012 [16] | T | 566 | RT+STS 80mg qd | 14D | ①②③ | Yes |
| | C | 201 | RT | | | |
| Li DG 2017 [17] | T | 48 | RT+STS 40mg qd | 10D | ①⑤ | Unclear |
| | C | 48 | RT | | | |
| Huang LY 2013 [18] | T | 40 | RT+STS 30mg qd | 14D | ① | Unclear |
| | C | 40 | RT | | | |
| Zhao FL 2012 [19] | T | 40 | RT+STS 40mg qd | 14D | ①⑤⑦ | No |
| | C | 40 | RT | | | |
| He F 2014 [20] | T | 30 | RT+STS 60mg qd | 15D | ①④⑦ | Unclear |
| | C | 30 | RT | | | |
| Zhang ZH 2013 [21] | T | 55 | RT+STS 60mg qd | 14D | ①④ | Yes |
| | C | 55 | RT | | | |
| Zou JM 2017 [22] | T | 140 | RT+STS 140mg qd | 28D | ①② | Unclear |
| | C | 140 | RT | | | |
| Du FD 2018 [23] | T | 50 | RT+STS 60mg qd | 14D | ①⑤ | Unclear |
| | C | 50 | RT | | | |
| Gao Y 2014 [24] | T | 30 | RT+STS 60mg qd | 14D | ①④⑥⑦ | Unclear |
| | C | 30 | RT | | | |
| Wang HY 2013 [25] | T | 50 | RT+STS 40mg qd | 28D | ①⑤⑦ | Unclear |
| | C | 50 | RT | | | |
| Chen G 2013 [26] | T | 40 | RT+STS 60mg qd | 28D | ①②③④ | Yes |
| | C | 40 | RT | | | |
| Fan ZJ 2016 [27] | T | 35 | RT+STS 60mg qd | 14D | ① | Unclear |
| | C | 35 | RT | | | |
| Zhang HY 2014 [28] | T | 40 | RT+STS 60mg qd | 14D | ①⑦ | No |
| | C | 40 | RT | | | |
| Li W 2012 [29] | T | 42 | RT+STS 40mg qd | 14D | ① | Yes |
| | C | 36 | RT | | | |
| Huang JN 2012 [30] | T | 60 | RT+STS 50mg qd | 14D | ①④⑥ | Unclear |
| | C | 60 | RT | | | |
| Pei X 2009 [31] | T | 36 | RT+STS 40mg qd | 14D | ①⑤ | Yes |
| | C | 35 | RT | | | |

(*Continued*)

**Table 1.** (Continued)

| References | Group | Sample size | Interventions | Duration | outcome indicators | adverse reactions |
|---|---|---|---|---|---|---|
| Wang JH 2010 [32] | T | 38 | RT+STS 60mg qd | 14D | ①④ | Unclear |
|  | C | 38 | RT |  |  |  |
| Ge HZ 2010 [33] | T | 30 | RT+STS 60mg qd | 15D | ①④⑥ | No |
|  | C | 30 | RT |  |  |  |
| Yang N 2010 [34] | T | 32 | RT+STS 60mg qd | 7D | ①④⑤⑦ | No |
|  | C | 32 | RT |  |  |  |
| Liu HL 2010 [35] | T | 50 | RT+STS 50mg qd | 14D | ① | No |
|  | C | 50 | RT |  |  |  |
| Huang H 2011 [36] | T | 53 | RT+STS 60mg qd | 14D | ①④ | No |
|  | C | 45 | RT |  |  |  |
| Luo WH 2011 [37] | T | 60 | RT+STS 50mg qd | 14D | ①⑤⑥ | No |
|  | C | 57 | RT |  |  |  |
| Song WJ 2011 [38] | T | 37 | RT+STS 40mg qd | 14D | ①④ | Unclear |
|  | C | 37 | RT |  |  |  |
| Zhang W 2011 [39] | T | 42 | RT+STS 40mg qd | 14D | ①④ | Unclear |
|  | C | 42 | RT |  |  |  |
| Chen XL 2008 [40] | T | 36 | RT+STS 50mg qd | 10D | ①②③④ | Yes |
|  | C | 28 | RT |  |  |  |
| Qin YW 2008 [41] | T | 35 | RT+STS 40mg qd | 15D | ①④⑦ | Yes |
|  | C | 34 | RT |  |  |  |
| Xu GP 2008 [42] | T | 37 | RT+STS 80mg qd | 7D | ⑦ | Unclear |
|  | C | 37 | RT |  |  |  |
| Wang QL 2007 [43] | T | 25 | RT+STS 50mg qd | 14D | ①④⑤⑦ | No |
|  | C | 25 | RT |  |  |  |
| Ma JH 2007 [44] | T | 30 | RT+STS 40mg qd | 14D | ①④ | No |
|  | C | 29 | RT |  |  |  |
| Zhang HR 2006 [45] | T | 26 | RT+STS 80mg qd | 14D | ①④ | Yes |
|  | C | 26 | RT |  |  |  |

(3)Time of angina attack:

4 studies [9,16,26,40] reported the onset time of angina pectoris, and the heterogeneity test showed significant differences among the studies (P < 0.00001, I² = 98%). The results of meta-

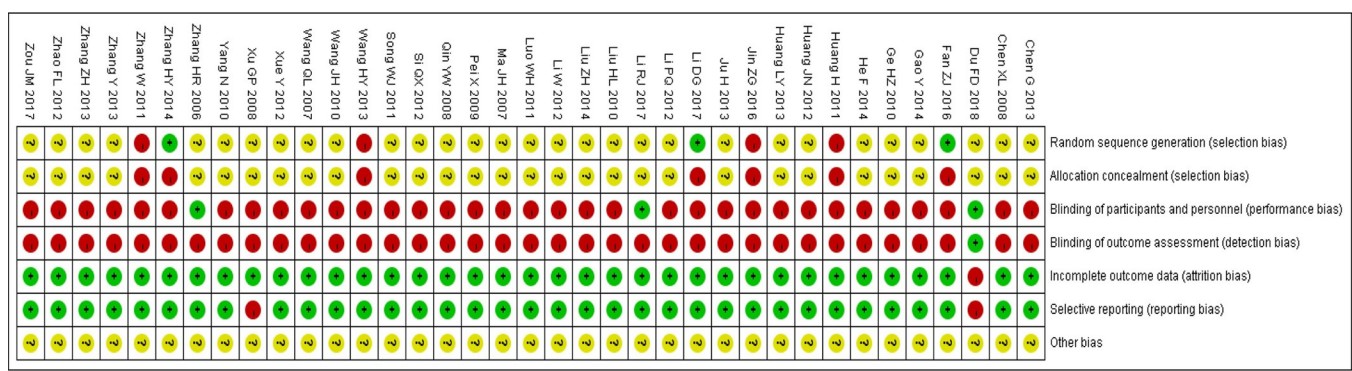

**Fig 2. Bias risk assessment for individual studies.**

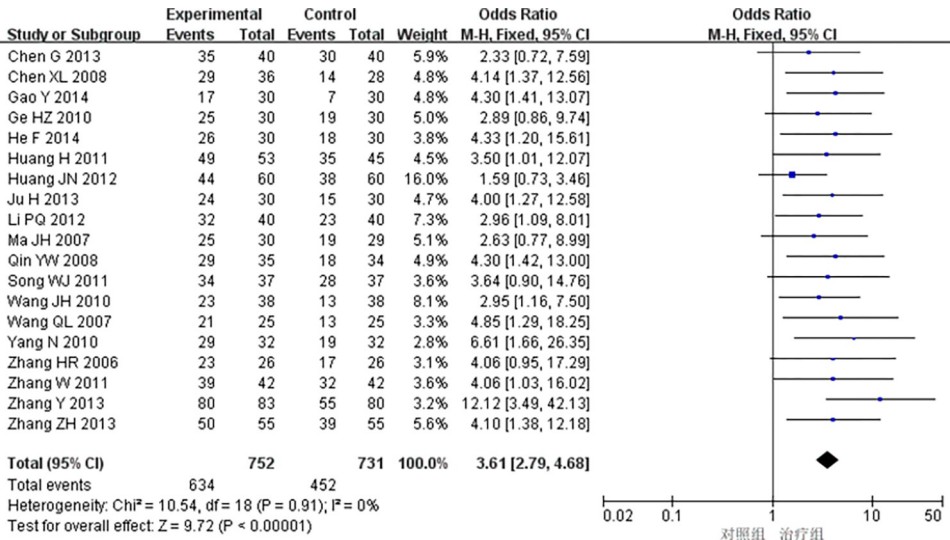

**Fig 3. Meta-analysis of ECG efficacy.**

analysis using a random effect model showed that the differences were statistically significant (SMD = -4.01, 95% CI [-6.18, -1.84], P = 0.0003). As shown in Fig 5, the treatment group received better efficacy in improving the duration of angina attack than the control group. Subgroup analysis of the treatment course was adopted to detect the cause of high heterogeneity, 2 articles [9,16] studied a treatment course of 14 days and found that the heterogeneity decreased significantly when the treatment course was 14 days (P = 0.95, $I^2$ = 0), with a statistically significant difference (SMD = -5.78, 95% Cl [-6.08, -5.48], P < 0.00001), indicating that STS combined with routine treatment was superior to the routine treatment alone in reducing the onset time of angina pectoris. In comparison, 2 studies [26,40] with a non-14-day treatment course reported significant heterogeneity (P = 0.002, $I^2$ = 89%), with statistically significant difference (SMD = -2.21, 95% Cl [-3.51, -0.92], P = 0.0008), indicating that STS combined with routine treatment has better efficacy in reducing the onset time of angina pectoris than the routine treatment alone. The subgroup analysis results show that the treatment effect is better when the course of treatment is set as 14 days, as shown in Fig 6.

**3.3.2 Secondary outcome indicators.** (1)Blood rheology

9 studies [9,17,19,23,25,31,34,37,43] reported hemorheology, and the heterogeneity test showed that there was a large heterogeneity among the studies ($P$<0.00001, $I^2$ = 90%). The random effects model was used for meta-analysis, and the results showed that the difference

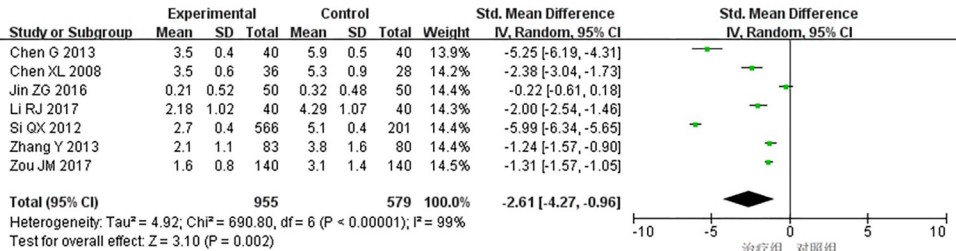

**Fig 4. Meta-analysis off requency of angina attacks.**

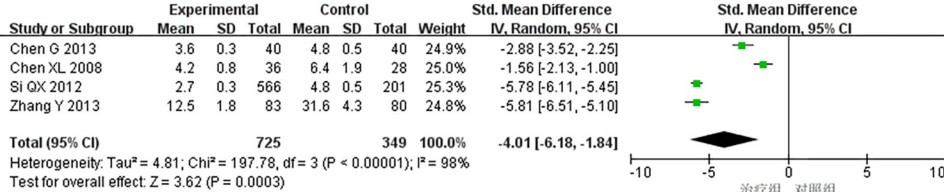

**Fig 5. Meta-analysis of angina pectoris attack time.**

was statistically significant (SMD = -0.93, 95%CI [-1.20, -0.66], $P$<0.00001), and the treatment group received better efficacy than the control group in improving hemorheology: ①Plasma viscosity: 9 studies [9,17,19,23,25,31,34,37,43] reported plasma viscosity with high heterogeneity ($P$<0.00001, $I^2$ = 93%), so random effects model was used for meta-analysis. The results showed that the difference was statistically significant(SMD = -1.02, 95%CI [-1.58, -0.47], $P$<0.0003), indicating that the treatment group received better efficacy than the control group in improving plasma viscosity; ②Whole blood low viscosity: 6 studies [9,17,19,25,31,37] reported whole blood low viscosity with high heterogeneity ($P$ = 0.0003, $I^2$ = 79%), and the random effects model was used for meta-analysis. The results showed that the difference was statistically significant(SMD = -0.85, 95%CI [-1.21, -0.49], $P$<0.00001), indicating that the treatment group received better efficacy than the control group in improving the whole blood low viscosity; ③ Whole blood high shear viscosity: 6 studies [9,17,19,25,31,37] reported whole blood high viscosity with high heterogeneity($P$<0.00001, $I^2$ = 91%), and the random effects model was used for meta-analysis. The results showed that the difference was statistically significant(SMD = -0.82, 95%CI [-1.44, -0.20], $P$ = 0.009), indicating that the treatment group received better efficacy than the control group in improving the whole blood high viscosity; ④Red blood cell aggregation: 5 studies [9,17,23,37,43] reported red blood cell aggregation with high heterogeneity($P$<0.00001, $I^2$ = 94%), and the random effects model was used for meta-analysis. The results showed that the difference was statistically significant(SMD = -1.00, 95%CI [-1.75, -0.25], $P$ = 0.009), indicating that the treatment group received better efficacy than the control group in improving the red blood cell aggregation Fig 7.

(2) Blood lipid

5 studies [13,24,30,33,37] reported blood lipid with high heterogeneity ($P$<0.00001, $I^2$ = 97%), and the random effects model was used for meta-analysis. The results showed that the

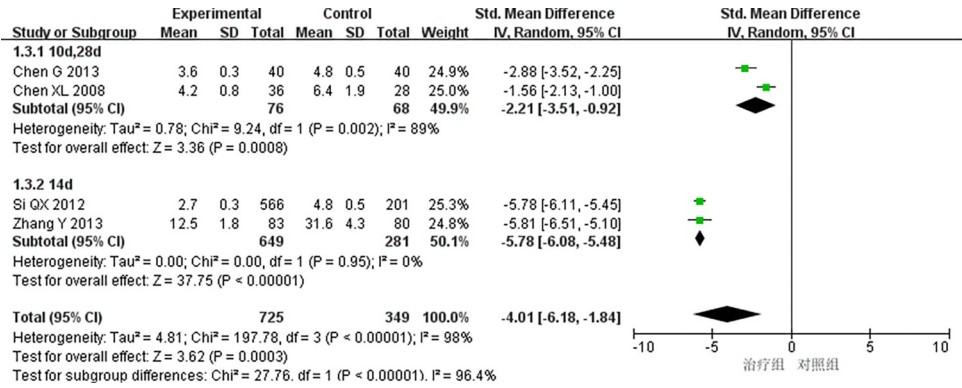

**Fig 6. Subgroup analysis of angina pectoris attack time.**

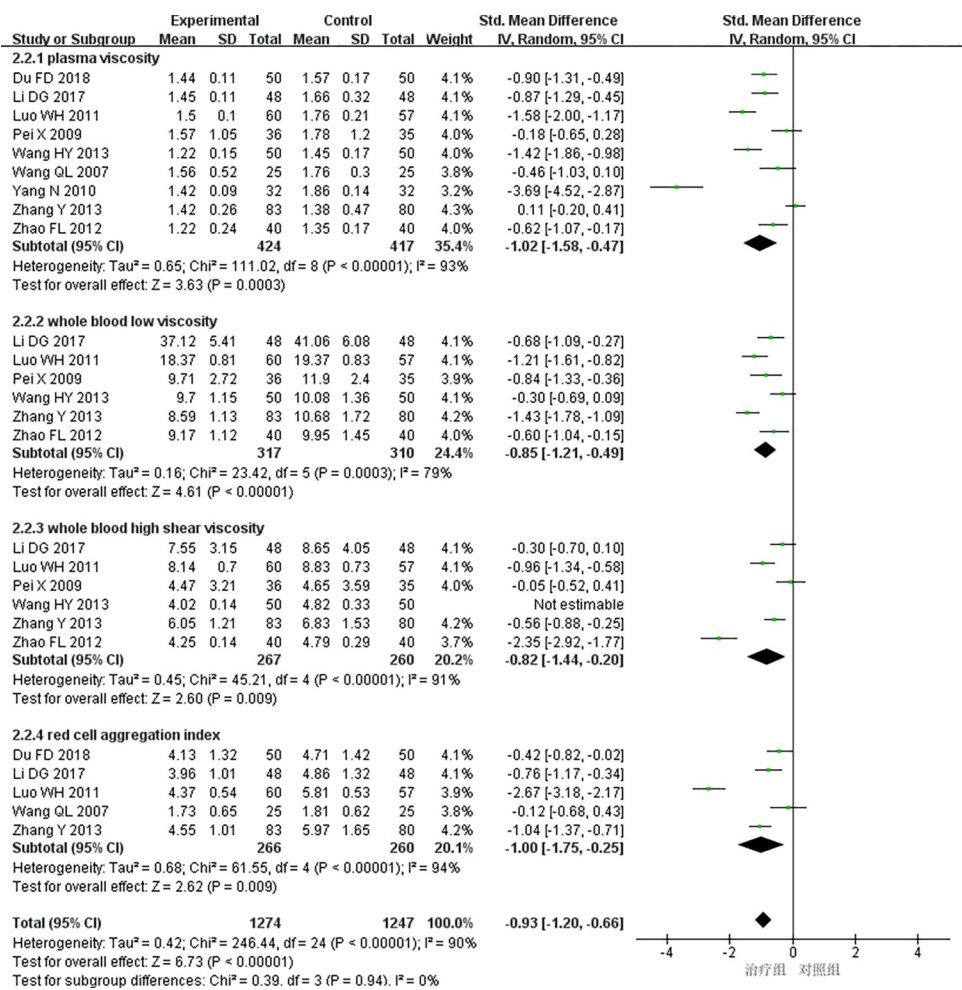

**Fig 7. Meta-analysis of blood rheology.**

difference was statistically significant (SMD = -1.15, 95%CI [-1.88, -0.42], *P* = 0.002), indicating that the treatment group received better efficacy than the control group in improving the blood lipid; ①TC: 4 studies [13,24,33,37] reported TC with high heterogeneity(*P*<0.00001, $I^2$ = 98%), and the random effects model was used for meta-analysis. The results showed that the difference was statistically significant(SMD = -3.86, 95%CI [-6.37, -1.34], *P* = 0.003), indicating that the treatment group received better efficacy than the control group in improving the TC; ②TG: 5 studies [13,24,30,33,37] reported TG with high heterogeneity(*P* = 0.0003, $I^2$ = 81%), and the random effects model was used for meta-analysis. The results showed that the difference was statistically significant (SMD = -0.60, 95%CI [-1.04, -0.16], *P* = 0.008), indicating that the treatment group received better efficacy than the control group in improving the TG; ③LDL-c: 4studies [13,24,30,33] reported LDL-c with high heterogeneity(*P*<0.00001, $I^2$ = 95%), and the random effects model was used for meta-analysis. The results showed that the difference was statistically significant(SMD = -1.54, 95%CI [-2.67, -0.42], *P* = 0.007), indicating that the treatment group received better efficacy than the control group in improving the LDL-c; ④HDL-C: 4 studies [13,24,30,33] reported HDL-c with high heterogeneity, and there was no significant difference between the treatment group and the control group (P = 0.14>0.05) Fig 8.

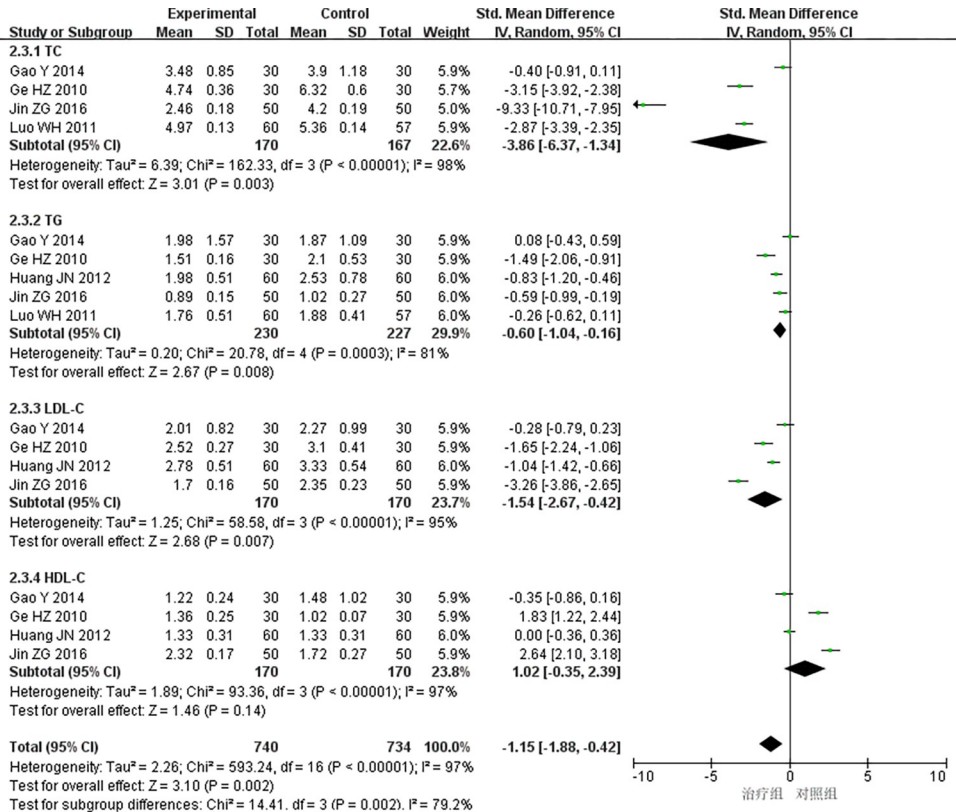

**Fig 8. Meta-analysis of blood lipid.**

(3) CRP

9studies [19,20,24,25,28,34,41–43] reported CRP with high heterogeneity ($P<0.00001$, $I^2 = 89\%$), and the random effects model was used for meta-analysis. The results showed that the difference was statistically significant(SMD = -1.39, 95%CI [-1.91, -0.86], $P<0.00001$), indicating that the treatment group received better efficacy than the control group in improving the CRP Fig 9.

**3.3.3 Adverse reactions.** 10 studies [9,14,16,21,26,29,31,40,41,45] reported adverse reactions. According to the heterogeneity test, the heterogeneity among the studies was small ($P = 0.56$, $I^2 = 0\%$), so fixed effects model was used for meta-analysis. The results showed that the difference was statistically significant (OR = 2.26, 95%CI [1.06, 4.85], $P = 0.04$), indicating

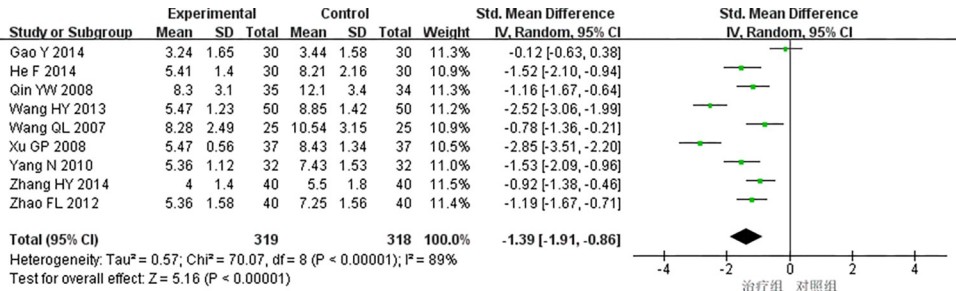

**Fig 9. Meta-analysis of CRP.**

**Table 2. Adverse reactions.**

| References | Treatment group | Control group |
|---|---|---|
| Zhang Y 2013 [9] | 2 cases had fever and 1 case had rash | None |
| Ju H 2013 [14] | 1 case had mild headache. | None |
| Si QX 2012 [16] | 5 cases of nausea, vomiting, occasional headaches | None |
| Zhang ZH 2013 [21] | 1 case of nausea, pain at the injection site. | None |
| Chen G 2013 [26] | None | 1 case had headache and red face discomfort. |
| Li W 2012 [29] | None | 3 cases of petechiae, ecchymoses subcutaneous injection site. |
| Pei X 2009 [31] | 2 cases of red face with dizziness | None |
| Chen XL 2008 [40] | 1 case of fever, 1 case of consciously headache, dizziness. | None |
| Qin YW 2008 [41] | 3 cases of dizziness, 2 cases of facial flush | None |
| Zhang HR 2006 [45] | 2 cases had swelling sensation in injection area | 1 case of slight headache |

that the incidence of adverse reactions in the treatment group was higher than that in the control group. However, the adverse reactions in the two groups were alleviated without treatment after the slowing down of the dripping rate or drug withdrawal, which did not affect the study process. There were no significant changes in liver and kidney function before and after treatment in the two groups. The specific adverse reactions are shown in Table 2 and Fig 10.

## 3.4 Sensitivity analysis

Sensitivity analysis was conducted based on the meta-analysis results of TST in the treatment of UAP. All outcome indicators were removed on a one-by-one basis and no significant changes in outcome indicators were found in the re-analysis, indicating low sensitivity of the overall analysis results and high confidence with the outcome.

## 3.5 Bias risk assessment

Risk of publication bias was assessed for outcome indicators of more than 10 articles included in this meta-analysis, and funnel plots of ECG efficacy and clinical symptom efficacy were observed. It was found that the study sites were not completely symmetrical, suggesting the risk of publication bias Figs 11 and 12.

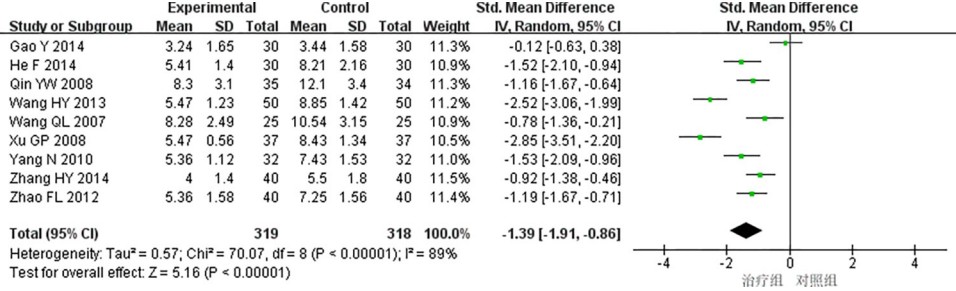

**Fig 10. Meta-analysis of adverse reactions.**

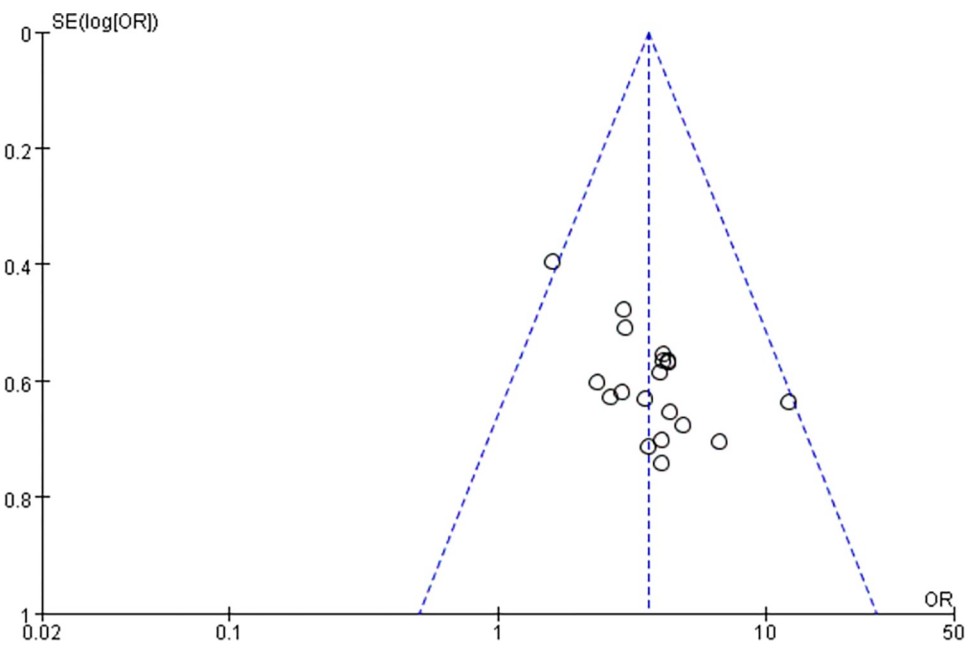

**Fig 11. ECG efficacy funnel chart.**

## 4. Discussion

### 4.1 Result analysis

Meta-analysis and systematic review have shown that the TST in combination with routine treatment can reduce the frequency and duration of angina attacks, improve ECG efficacy and clinical symptoms, reduce TG, TC, LDL-C, lower whole blood low shear viscosity, whole

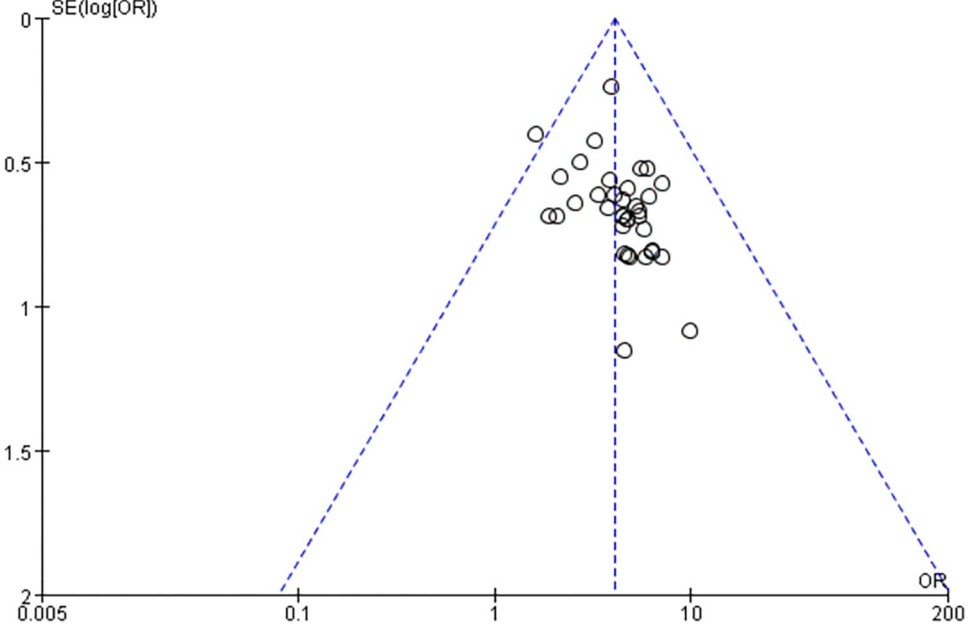

**Fig 12. Clinical efficacy funnel chart.**

blood high shear viscosity, plasma viscosity as well as red blood cell aggregation. The effect of CRP reduction was superior to that of routine treatment alone. As for the improvement of HDL-C, no ideal results were obtained due to the inadequate quantity and uneven quality of the included studies and further confirmation was needed. Subgroup analysis showed that 14 days of treatment was more effective in reducing the duration of angina attack. A few cases of adverse reactions occurred in the two groups, and the incidence of adverse reactions in the treatment group was higher than that in the control group. However, no abnormal liver and kidney function occurred in each study, which did not affect the research process. The clinical application of TST in the treatment of UAP is safe and effective, but the quality of the included literature is low. Therefore, higher quality randomized controlled clinical trials are needed in the future to further demonstrate the effectiveness and safety of TST and provide better clinical guidance.

### 4.2 Limitations

This study systematically evaluated the efficacy and safety of STS in combination with routine treatment for UAP, with the following limitations: ① Of the 37 articles included, only 2 were randomized with numerical tables, 1 was randomized with a computer-generated program, 1 was randomized in the order of patient visits, 3 were randomized in the order of admission, and remainder just mentioned randomization, without specifying which randomization method was adopted. Only 2 articles used a single-blind experimental design. ② The quality of the included literature is uneven, and the heterogeneity in age, sex, intervention course, and dose may also be important factors leading to biases. ③ There was a lack of long-term follow-up study and therefore an absence of long-term observation data.

### 4.3 Clinical significance

STS is mainly used in the clinical treatment of cardiovascular and cerebrovascular diseases, such as coronary heart disease and angina pectoris [46], which can effectively improve angina pectoris related indexes, optimize hemorheology indexes, improve blood lipid and reduce CRP level [47]. The significance of this systematic review mainly lies in the comprehensive collection of all Chinese and English literatures on STS combined with routine treatment in the UAP treatment. After data screening and meta-analysis, a large-sample study result was finally obtained, which is more convincing and reliable than single-sample study [48].

In order to further strengthen the evidence, future studies should develop rigorous implementation plans, design large-sample, multi-center and long-term clinical randomized controlled trials according to evidence-based medicine standards, and pay attention to the correct and reasonable implementation of randomization, allocation concealment and blind methods, so as to reduce the biases in selectivity, implementation, measurement and other aspects. For the treatment of UAP by STS, different doses and courses of medication can produce different curative effects on patients. Normative standards of dosage and course of treatment should be formulated in the included studies according to the use method of STS instructions and suitable population. In addition, the rate of drops should be controlled, with special attention paid to the allergic population, so as to improve the active treatment of adverse reactions in the treatment process. Records and reports should be made in a more detailed manner to improve the quality of evidence [49].

## 5. Conclusion

In conclusion, based on the available evidence, the STS in combination with routine treatment may be promisingly effective and safe in the treatment of UAP However, the overall quality of

the included literature is low, and higher quality clinical randomized controlled trial are needed in the future to further explore the effectiveness and safety of STS, so as to provide higher quality evidence for clinical work.

## Supporting information

**S1 Checklist. PRISMA 2009 checklist.**
(DOC)

## Author Contributions

**Conceptualization:** Xiaoqi Wu, Dong Guo.

**Data curation:** Xiaoqi Wu, Maoxia Fan.

**Formal analysis:** Sining Wei.

**Investigation:** Maoxia Fan.

**Methodology:** Maoxia Fan.

**Software:** Xiaoqi Wu.

**Supervision:** Dong Guo.

**Validation:** Dong Guo.

**Visualization:** Sining Wei.

**Writing – original draft:** Xiaoqi Wu.

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
