## [Decision Letter · Decision Letter 0]

22 May 2023

PONE-D-23-09981The efficacy and safety of Sodium tanshinone ⅡA sulfonate injection in the treatment of unstable angina pectoris: a systematic review and meta-analysisPLOS ONE

Dear Dr. Guo,

Thank you for submitting your manuscript to PLOS ONE. After careful consideration, we feel that it has merit but does not fully meet PLOS ONE’s publication criteria as it currently stands. Therefore, we invite you to submit a revised version of the manuscript that addresses the points raised during the review process.

Reviewer 1

This long article over 6,500 words as presently written and presented cannot be accepted because:

1, it is written in poor English with numerous grammatical and spelling mistakes

2. most of the original articles are written in Chinese (36) and cannot be accessed by the interested reader without translation resources. Many of the articles appear to be from small local journals rather than Chinese national journals.

3. Table 1 is poorly presented, with articles written by name of first author and year of publication WITHOUT REFERENCE Number, making it difficult to check on data. In general, most tables and figures contain too much data and are poorly labelled, making them difficult to read.

5. the Forest plots are favoring treatment vs control are written in Chinese

6. the presentation of the effects of treatment is poorly organized, so that the reader does not know whether authors want to highlight clinical, ECG or laboratory effects of treatment.

This is actually an important topic, and there have been previous smaller metanalyses and trials on the same subject showing efficacy of STS in angina. Authors have put in quite a lot of effort in this study but have written findings poorly and in a manner that cannot be published.

Suggestions:

They should seek assistance of a colleague with writing experience in English language international medical publications to rewrite this article.

They should omit minor unimportant issues and organize the results under clinical effects and ECG/laboratory effects of treatment.

They should not add in analysis of the quality of studies (Table 2, Fig 2 and 3) as doing so overloads the reader and distracts from their results.

They should not have a separate Fig 8 (which is shown TWICE) on symptoms, since effect on angina is already reported in detail in Figures 5, 6 and 7.

They should not present GRADE analysis since all data appear to be of low or very low quality and again this distracts from the large amount of information they have already shown.

Reviewer 2

The meta analysis is conducted in a proper way and follow the guide for the meta analysis format. The conclusion for the manuscript may alter to "based on the available evidence, the combination of STS with routine treatment may be promisingly effective and safe in the treatment of UAP" as the funnel chart showed asymmetry that may affect the consistency of the statement of conclusion. There are quite a number of spelling errors that need to be corrected. The detail may refer to the attached file.

We look forward to receiving your revised manuscript.

Kind regards,

Hean Teik Ong

Academic Editor

PLOS ONE

Journal Requirements:

4. Please clarify the Figure 8 "Meta analysis of efficacy of clinical symptomsy" in page "21" and Figure 8 "Meta analysis of efficacy of clinical symptomsy" in page "23".

5. Please upload a copy of Figure 9, to which you refer in your text on page 22. If the figure is no longer to be included as part of the submission please remove all reference to it within the text.

6. Please remove your figures from within your manuscript file, leaving only the individual TIFF/EPS image files, uploaded separately. These will be automatically included in the reviewers’ PDF.

Reviewers' comments:

Reviewer's Responses to Questions

**Comments to the Author**

1. Is the manuscript technically sound, and do the data support the conclusions?

Reviewer #1: Partly

Reviewer #2: Yes

2. Has the statistical analysis been performed appropriately and rigorously? 

Reviewer #1: I Don't Know

Reviewer #2: Yes

3. Have the authors made all data underlying the findings in their manuscript fully available?

Reviewer #1: No

Reviewer #2: Yes

4. Is the manuscript presented in an intelligible fashion and written in standard English?

Reviewer #1: No

Reviewer #2: No

5. Review Comments to the Author

Reviewer #1: This long article over 6,500 words as presently written and presented cannot be accepted because:

1, it is written in poor English with numerous grammatical and spelling mistakes

2. most of the original articles are written in Chinese (36) and cannot be accessed by the interested reader without translation resources. Many of the articles appear to be from small local journals rather than Chinese national journals.

3. Table 1 is poorly presented, with articles written by name of first author and year of publication WITHOUT REFERENCE Number, making it difficult to check on data. In general, most tables and figures contain too much data and are poorly labelled, making them difficult to read.

5. the Forest plots are favoring treatment vs control are written in Chinese

6. the presentation of the effects of treatment is poorly organized, so that the reader does not know whether authors want to highlight clinical, ECG or laboratory effects of treatment.

This is actually an important topic, and there have been previous smaller metanalyses and trials on the same subject showing efficacy of STS in angina. Authors have put in quite a lot of effort in this study but have written findings poorly and in a manner that cannot be published.

Suggestions:

They should seek assistance of a colleague with writing experience in English language international medical publications to rewrite this article.

They should omit minor unimportant issues and organize the results under clinical effects and ECG/laboratory effects of treatment.

They should not add in analysis of the quality of studies (Table 2, Fig 2 and 3) as doing so overloads the reader and distracts from their results.

They should not have a separate Fig 8 (which is shown TWICE) on symptoms, since effect on angina is already reported in detail in Figures 5, 6 and 7.

They should not present GRADE analysis since all data appear to be of low or very low quality and again this distracts from the large amount of information they have already shown.

Reviewer #2: The meta analysis is conducted in a proper way and follow the guide for the meta analysis format. The conclusion for the manuscript may alter to "based on the available evidence, the combination of STS with routine treatment may be promisingly effective and safe in the treatment of UAP" as the funnel chart showed asymmetry that may affect the consistency of the statement of conclusion. There are quite a number of spelling errors that need to be corrected. The detail may refer to the attached file.

6. PLOS authors have the option to publish the peer review history of their article (what does this mean?). If published, this will include your full peer review and any attached files.

Reviewer #1: **Yes: **Hean Teik Ong

Reviewer #2: No

---

## [Author Response · Author response to Decision Letter 0]

28 Jul 2023

Dear Editor,

Thank you for your letter and for the editor’s comments concerning our manuscript entitled “The efficacy and safety of sodium tanshinone ⅡA sulfonate injection in the treatment of unstable angina pectoris: a systematic review and meta-analysis” (Manuscript Number: PONE-D-23-09981).

Those comments are all valuable and very helpful for revising and improving our paper, as well as the important guiding significance to our research. We have studied comments carefully and have made a correction which we hope meet with approval. Revised portions are marked in red on the paper. The main corrections in the paper and the response to the editor’s comments are as flowing: 

Responds to the reviewers comments: 

1.Response to reviewer 1 comment:

- 1. It is written in poor English with numerous grammatical and spelling mistakes.

Response: Thank you for pointing out this point. I have corrected language and grammatical errors, and seek colleague with writing experience in English language international medical publications to rewrite this article.

2.Response to reviewer 1 comment:

- 2. They should omit minor unimportant issues and organize the results under clinical effects and ECG/laboratory effects of treatment.

Response: Thank you for pointing out this point. We have added links to the references mentioned in the table, comprehensively representing some indicators. The results emphasize clinical and laboratory efficacy, making it easier to read.

3.Response to reviewer 1 comment:

- 3. They should not add in analysis of the quality of studies (Table 2, Fig 2 and 3) as doing so overloads the reader and distracts from their results.

Response: Thank you for pointing out this point. Meta analysis as a data consolidation method of quantitative Systematic review, the quality of its conclusions depends not only on the strict operational process of meta analysis, but also on the quality of the research literature itself and its control of bias. Therefore, the author believes that it is necessary to evaluate the quality and bias of the research literature included in the meta-analysis. However, considering the focus and readability of this article, we removed Table 2 and Fig 2.

4.Response to reviewer 1 comment:

-4. They should not have a separate Fig 8 (which is shown TWICE) on symptoms, since effect on angina is already reported in detail in Figures 5, 6 and 7.

Response: Thank you for pointing out this problem. Due to our negligence in uploading Figure 8 over Figure 9 (Meta-analysis of blood rheology), we have removed Figure 8 from the article and added Figure 9 as suggested.

5.Response to reviewer 1 comment:

-5. They should not present GRADE analysis since all data appear to be of low or very low quality and again this distracts from the large amount of information they have already shown.

Response: Thank you for pointing out this problem. We have removed G from this article.

6.Response to reviewer 2 comment:

-6. The conclusion for the manuscript may alter to "based on the available evidence, the combination of STS with routine treatment may be promisingly effective and safe in the treatment of UAP" as the funnel chart showed asymmetry that may affect the consistency of the statement of conclusion. 

Response: Thank you for pointing out this problem. We have revised the wording of the conclusion according to your suggestion.

7.Response to reviewer 2 comment:

-7. There are quite a number of spelling errors that need to be corrected.

Response: Thank you for pointing out this problem. I have corrected the language and grammatical errors. See the red marks in the paper for details.

8.Response to Academic Editor comment:

-8. Please ensure that your manuscript meets PLOS ONE's style requirements, including those for file naming.

Response: Thank you for pointing out this problem. We have made changes to the manuscript format as required to ensure that it meets the style requirements of PLOS ONE.

9.Response to Academic Editor comment:

-9. We note that the grant information you provided in the‘Funding Information’and‘Financial Disclosure’ sections do not match. 

Response: Thank you for pointing out this problem.I have provided the correct grant numbers for the awards received for research in the 'Funding Information' section.

10.Response to Academic Editor comment:

-10. PLOS requires an ORCID iD for the corresponding author in Editorial Manager on papers submitted after December 6th, 2016. Please ensure that you have an ORCID iD and that it is validated in Editorial Manager. 

Response: Thank you for pointing out this problem.ORCID iD has been conducted and verified in Editorial Manager.

11.Response to Academic Editor comment:

-11. Please clarify the Figure 8 "Meta analysis of efficacy of clinical symptomsy" in page "21" and Figure 8 "Meta analysis of efficacy of clinical symptomsy" in page "23".

Response: Thank you for pointing out this problem.According to the opinions of the external audit experts, I have deleted Figure 8.

12.Response to Academic Editor comment:

-12. Please upload a copy of Figure 9, to which you refer in your text on page 22. If the figure is no longer to be included as part of the submission please remove all reference to it within the text.

Response: Thank you for pointing out this problem.I have uploaded the original Figure 9, now Figure 7 Meta-analysis of blood rheology. 

13.Response to Academic Editor comment:

-13. Please remove your figures from within your manuscript file, leaving only the individual TIFF/EPS image files, uploaded separately. These will be automatically included in the reviewers’PDF.

Response: Thank you for pointing out this problem.We have followed your suggestion to upload.

We tried our best to improve the manuscript and made some changes in the manuscript. And here we did not list the changes but marked them in red in the revised paper. We appreciate for Editor’s warm work and hope that the correction will meet with approval.

Once again, thank you very much for your comments and suggestions.

Sincerely

Xiaoqi Wu

---

## [Decision Letter · Decision Letter 1]

17 Aug 2023

The efficacy and safety of Sodium tanshinone ⅡA sulfonate injection in the treatment of unstable angina pectoris: a systematic review and meta-analysis

PONE-D-23-09981R1

Dear Dr. Dong Guo,

We’re pleased to inform you that your manuscript has been judged scientifically suitable for publication and will be formally accepted for publication once it meets all outstanding technical requirements.

Kind regards,

Hean Teik Ong

Academic Editor

PLOS ONE

Additional Editor Comments (optional):

PLEASE MODIFY CONCLUSION AS SUGGESTED BY REVIEWER.

Reviewers' comments:

Reviewer's Responses to Questions

**Comments to the Author**

1. If the authors have adequately addressed your comments raised in a previous round of review and you feel that this manuscript is now acceptable for publication, you may indicate that here to bypass the “Comments to the Author” section, enter your conflict of interest statement in the “Confidential to Editor” section, and submit your "Accept" recommendation.

Reviewer #2: All comments have been addressed

Reviewer #3: All comments have been addressed

2. Is the manuscript technically sound, and do the data support the conclusions?

Reviewer #2: Yes

Reviewer #3: Partly

3. Has the statistical analysis been performed appropriately and rigorously? 

Reviewer #2: Yes

Reviewer #3: Yes

4. Have the authors made all data underlying the findings in their manuscript fully available?

Reviewer #2: Yes

Reviewer #3: Yes

5. Is the manuscript presented in an intelligible fashion and written in standard English?

Reviewer #2: Yes

Reviewer #3: Yes

6. Review Comments to the Author

Reviewer #2: Author has improved the writing and arrangement of the manuscript presentation. Most comments have been addressed by the author too.

Reviewer #3: Due to the low quality of the data, the conclusion of definite clinical efficacy should not be made. Please rephrase the conclusion.

7. PLOS authors have the option to publish the peer review history of their article (what does this mean?). If published, this will include your full peer review and any attached files.

Reviewer #2: **Yes: **Tan Boon Seng

Reviewer #3: No

---

## [Editor Report · Acceptance letter]

23 Aug 2023

PONE-D-23-09981R1 

The efficacy and safety of sodium tanshinone ⅡA sulfonate injection in the treatment of unstable angina pectoris: a systematic review and meta-analysis 

Dear Dr. Guo:

I'm pleased to inform you that your manuscript has been deemed suitable for publication in PLOS ONE. Congratulations! Your manuscript is now with our production department. 

Kind regards, 

on behalf of

Dr. Hean Teik Ong 

Academic Editor

PLOS ONE